# The Effects of a Scenario-Based Spiritual Care Course on Spiritual Care Competence among Clinical Nurses: A Quasi-Experimental Study

**DOI:** 10.3390/healthcare11010036

**Published:** 2022-12-22

**Authors:** Suh-Ing Hsieh, Li-Ling Hsu, Katherine A. Hinderer, Hui-Ling Lin, Yi-Ping Tseng, Chen-Yi Kao, Ching-Yun Lee, Shu-Hua Kao, Yen-Fang Chou, Li-Yun Szu, Lun-Hui Ho

**Affiliations:** 1Department of Nursing, Chang Gung University of Science and Technology, Taoyuan City 33303, Taiwan; 2Department of Nursing, Taoyuan Chang Gung Memorial Hospital, Taoyuan City 33378, Taiwan; 3Ching Kuo Institute of Management & Health, Keelung 203301, Taiwan; 4Institute for Nursing Research and Evidence-Based Practice, Connecticut Children’s Medical Center, Hartford, CT 06106, USA; 5Department of Pediatrics, School of Medicine, University of Connecticut, Farmington, CT 06030, USA; 6Department of Nursing, Linkou Chang Gung Memorial Hospital, Taoyuan City 33375, Taiwan; 7Department of Nursing, College of Nursing, Taipei Medical University, Taipei City 110, Taiwan; 8School of Nursing, College of Medicine, National Taiwan University, Taipei City 100, Taiwan; 9Hospice and Palliative Care Ward, Taoyuan City 33353, Taiwan; 10Department of Nursing Management of the Administration Center, Chang Gung Medical Foundation, Taoyuan City 33375, Taiwan; 11Department of Nursing, Chiayi Chang Gung Memorial Hospital, Puzi City 613, Taiwan

**Keywords:** simulation, spiritual care competency, clinical nurse, objective structure clinical examination, standardized patient

## Abstract

Across their lifespans, and in many clinical settings, patients have spiritual care needs. Many nurses lack competence related to providing spiritual care. Popular educational strategies, such as simulated educational programs and objective structured clinical examinations (OSCE), have not been widely adopted in nursing spiritual care education. The purpose of this study was to explore the effects of a scenario-based spiritual care course on spiritual care competence in nurses. This quasi-experimental study employed a repeated-measures pre-test/post-test design with assessments immediately before, immediately after, and 3 months post-intervention. Nurses providing direct patient care in diverse clinical settings were recruited from a large medical foundation in northern Taiwan. The intervention was a one day scenario-based spiritual care course and OSCE. The experimental group (*n* = 53) and controls (*n* = 85) were matched for their similar units, ages, working experience, and clinical ladder status. The Spiritual Care Competence Scale (SCCS), Spiritual Perspective Scale (SPS), Spiritual Care Perspective Scale-Revised (SCPS-R), and reflection logs were completed by both experimental and control groups. The Course Satisfaction Scale, OSCE Checklist, and Standardized Patient Feedback Scale (SPFS) were completed by the experimental group only. The experimental group had significantly higher SPS scores and self-evaluated SCCS scores, and lower SCPS-R scores (more positive spiritual perspectives), than controls at 3 months post-intervention. The experimental group showed significant within-subject effects at three time points on SPS scores, SCPS-R scores, and self-evaluated SCCS scores. Mean global performance of OSCE was 3.40 ± 0.91, and SP feedback indicated strengths and areas for improvement. In conclusion, the scenario-based spiritual care course effectively enhanced nurses’ spiritual care competence, competence, and skills. Blended education techniques can therefore enhance nurses’ ability to support patients with spiritual care needs.

## 1. Introduction

### 1.1. Background

The International Council of Nurses (ICN) [1] defines nursing as a combination of health promotion, illness prevention, and direct care of sick, disabled, and dying people across all ages, families, groups, and communities in all settings [1]. As patients and their families experience stress, aging, illness, disability, and/or dying, unmet spiritual needs are often provoked. Spirituality fulfills the meaning of life, fosters hope for survival, leads to a peaceful mindset, and preserves one’s dignity. Spiritual needs may encompass experiencing reciprocal human love and receiving assistance to face death peacefully, which is actualized through caring and respect, caring for the mortal body, and transcending the worldly being [2,3,4,5]. Nurses support the spiritual needs of aging, ill, disabled, or dying people and their families, allowing them to reflect on self-esteem and to consider life’s meaning and purpose while providing emotional comfort and instilling hope [6]. Evidence has shown that spirituality is associated with better physical and mental health in patients [7,8]. Integrating spiritual care into nursing practice is supported by international accreditation and professional nursing organizations [9]. For instance, the Joint Commission recognizes the importance of spiritual care in healthcare [10]. The American Nurses’ Association integrates spirituality into the Scope and Standards of Nursing Practice. Standards that include spiritual care are competent in collecting spiritual/transpersonal data (Standard 1), developing an individualized spiritual plan (Standard 4), and using health promotion and health teaching methods appropriate to the healthcare consumer’s spirituality (Standard 5B) [11]. Finally, the ICN posits that, “nurses promote an environment in which religious and spiritual beliefs of the individual, families, and communities are acknowledged and respected by everyone” [12] (p.7).

However, the cultivation of spiritual care practices has not been actualized in either entry-level nursing educational preparation or clinical nurse continuing education [13,14,15]. Studies in Malta, Turkey, Iran, the United States, and Taiwan have shown that nursing education and clinical in-service education does not address spiritual care [13,16,17,18,19,20]. Clinical nurses often overlook spiritual care or refer the patients to religious personnel. This may be the result of factors at an individual level (viewing spirituality as meeting patients’ religious needs rather than an integrated element of nursing practice, for example), professional education levels (lack of nursing education on the meaning of spirituality and its differentiation from religion), and organizational level (a medical model of care delivery and a business model of managing the healthcare system) factors [21,22,23,24,25].

While nurses encounter spiritual concerns and issues of patients and families across their lifespan in a multitude of clinical settings, many lack spiritual care competence. The capacity of the nurses to assess for and provide interventions to care for patients’ spiritual needs along with needed knowledge, skills, and attitudes for delivery spiritual care is the definition of spiritual care competence [26]. The dimensions of the spiritual care competence scale (SCCS) are: “assessment and implementation of spiritual care, professionalisation and improving the quality of spiritual care, personal support and patient counselling, referral to professionals, attitude towards patient spirituality, and communication” [26]. Nursing education has shifted from acquiring information via didactic methods to using experiential learning to improve clinical nurse problem solving [27]. In contrast, the clinical care setting continues to use competency-based nursing education with competency-based assessments to evaluate learners’ proficiency [28]. 

Although two systematic reviews of spiritual care training for healthcare professionals were published [25,29], only six studies (seven publications) addressed spiritual care-related educational programs for clinical nurses. Of these six studies, one was a quasi-randomized controlled trial [30] and one was a two-group quasi-experimental study [31], Three studies employed a single group pre-test, post-test design [9,32,33]. The final study was a single group pre-test post-test repeated-measures design [34,35]. In five of the six studies, nurse participants were recruited from pediatric, hospice, and cancer wards [9,30,31,32,33,34,36], whereas only one of the six recruited nurses came from multiple settings, including an internal medicine ward, neurology ward, cardiology ward, coronary care unit, and mixed pulmonary disease/urology ward [31]. Didactic lecture was the primary teaching method in all of these studies. The authors of this paper were unable to find any nursing spirituality studies that used scenario simulation and objective structured clinical examination (OSCE) with standardized patients (SPs) as educational methods. Therefore, this study aimed to validate the effects of a scenario-based spiritual care courses on spiritual care competence among clinical nurses.

### 1.2. Program Design

Nurses may cultivate spiritual care competence through educational programs that integrate evidence-based learning strategies. Thus, the educational program in this study was based on the spiritual care educational model of the Action Spirituality and Spiritual Care in Education (ASSET) program [35,37], along with situational simulation [38,39,40,41,42,43,44,45] and objective structured clinical examinations (OSCEs) [46,47,48]. The ASSET program consists of three elements that include structure and content containing self-awareness and spiritual dimension in spirituality and nursing, they are as follows: (1) self-awareness is necessary before understanding the spirituality of others. Self-exploration enables nurses to examine themselves, their spiritual beliefs and values and clarifies self-meaning, increasing their sensitivity toward the spiritual needs of others. (2) Spirituality is the essence of existence and provides meaning and purpose in one’s personal existence. It is the intangible dimension of connecting others to their surroundings. Self-exploration and spiritual concepts can be integrated into the beginning of the educational program to assist participants in clarifying their values. (3) The spiritual dimension in nursing involves first exploring self-awareness and spirituality, which then allows new knowledge, attitudes, and skills to be applied to one’s spiritual dimension in nursing [35,37].

## 2. Materials and Methods

### 2.1. Study Design

This study employed a quasi-experimental pre-test post-test repeated measures design (Table 1). 

### 2.2. Participants and Setting

A convenience sample of direct care nurses was recruited from three branches of a large medical foundation in northern Taiwan. Recruitment methods included the use of recruitment posters, with QR codes, placed on multiple wards at three different times. 

Clinical nurses that provided direct continuous patient care in different hospital departments for at least 8 h per day, several days per week; nurses and head nurses from outpatient settings, health checkups, emergency department, operating room, dialysis units, and chemotherapy room, and those that provided task-oriented and non-continual patient care for less than 8 h per day, were excluded.

The control group was identified through snowball sampling. Experimental group study participants were asked to identify potential control group participants from their respective units. Control group participants were matched for similar units, ages, working experience, and nursing clinical ladder with the experimental group by a 1:1–3 ratio for considering the number of experimental participants, the number of nurses at different wards, and required sample size of the control group.

Head nurse (HN) participants were recruited based upon those who consented to participate in treatment or control groups. Head nurses were asked to complete spiritual care assessments of their respective nurses who were study participants. This provided an additional set of data with regard to the spiritual care offered by each nurse participant in the clinical setting. This data provided a more subjective view of the nurses’ spiritual care practices.

There were no published studies with effect size of between-subjects effects that used the study instruments to estimate the sample size [8,30,31,32,36,49]. In addition, recruiting clinical nurses for a study of this nature is difficult, thus it was not feasible to conduct a pilot study to determine the effect size. Thus, a power calculation of an unpaired *t* test with two-tailed, medium effect size 0.5, α 0.05, power 0.8, and 10% attrition via G*Power 3.1 was used to estimate the sample size of 136. However, the post hoc power analyses of between-subjects effects using multiple linear regression was between 0.999 (HN-evaluated SCCS) and 1.000 (SPS, SCPS-R, & self-evaluated SCCS).

### 2.3. Intervention

The eight learning objectives and curriculum of the scenario-based spiritual care course were crafted as the team considered the published evidence, ANA’s Scope and Standards, the ASSET program, and personal clinical practice experiences. Each element of the course was designed to address specific content that was central to understanding spiritual care in nursing (Table 2) [11,21,26,35,36,37,43,49,50,51,52]. With consideration for the learning objectives, the content of the course contained the spiritual impact of illness; definitions of spirituality, religion, and spiritual care; talking about spiritual clues of patients and families; spiritual needs assessment; spiritual care nursing process; spiritual distress and wellbeing; spiritual care methods and skills; searching life’s meaning and purpose, and individual spiritual reflection [13,22,31,53,54,55,56]. Six experts on spiritual care practice, hospice and palliative care, and simulation were invited to evaluate the PowerPoint content and the spiritual care scenario using a 4-point Likert scale. The content validity of index (CVI) was 1.00. For the OSCE portion of the program, the content validity of the scenario, template, and checklist, and SP feedback scale were reviewed by six experts using a 4-point Likert scale. The content validity index (CVI) of the OSCE scenario and template was 1.00.

The four-hour scenario-based spiritual care course was offered on 10 different dates (multiple sessions on some dates) for a total of 15 sessions in order to meet the scheduling needs of the study participants (Table 2). The number of participants per session ranged from one to 11 nurses and courses were offered between 17 August 2019 and 28 November 2020. Each offering of the course was facilitated by a qualified hospice and palliative care ward head nurse [(HN) HN session facilitator] who was trained by the principal investigator (PI, S-I.H) and had experience with teaching/learning strategies and simulation. The head nurse (HN) also served as examiner for the course and completed measures to assess the experimental group participants. In addition, standardized patients (SPs) were trained by the principal investigator prior to the course offering. The course was held in the Clinical Skills Center at Linkou Chang Gung Memorial Hospital, Taoyuan, Taiwan.

The initial segment of the course (morning session) consisted of a spiritual care lecture with a handout of relevant materials followed by a scenario-based video and debriefing by the HN session facilitator. The scenario-based video was a case of a 30-year-old female with palpitations admitted for cardiac catheterization (Table 2). In the second half of the course (afternoon session), the OSCE portion of the program occurred. Each participant attended a station of the OSCE examination, a 19-year-old cervical cancer patient, for 15 min and received SP feedback for 5 min each time. OSCE examiners were trained by the principal investigator and watched good and poor versions of a video two days before viewing the OSCE examination to assess inter-rater reliability. Additionally, examiners were trained on how to assess the performance of each participant. Standardized patients were experienced actors and were trained by the PI before the OSCE session. At program completion, experimental group participants were given a 500-New Taiwan dollar gift card as a token of appreciation for their time and 3 h of continuing education credits. 

### 2.4. Instruments

Outcomes data were collected via a variety of instruments in an effort to assess spiritual care outcomes (Table A1). In addition, an investigator-developed background information measure assessed demographic data from participants at the first time point of data collection (Table 1). Primary outcomes were measured by the Spiritual Perspective Scale (SPS), Spiritual Care Perspective Scale-Revised (SCPS-R), self- and head nurse-evaluated Spiritual Care Competence Scale (SCCS), and reflective logs. The Standardized Patient Feedback Scale (SPFS), OSCE Checklist, and Course Satisfaction Scale (CSS) measured the secondary outcomes. 

The developers, operational definitions, validity, and reliability of the included instruments are shown in Table A1. English-language versions of the SPS and SCPS-R were translated into Chinese, and the Chinese versions of both scales were translated into English by two English educators. Then, seven (SPS, SCPS-R, SCCS in September 2017) and six experts (reflective logs, SPFS, OSCE Checklist, & CSS in May 2019) on spiritual care practice, hospice and palliative care, and a Buddhist chaplain in the clinical setting were invited to assess these tools using a 4-point Likert scale. The final versions of these scales were revised based on experts’ comments and consideration of Chinese culture.

### 2.5. Data Collection

This study was conducted between 17 August 2019 and 28 February 2021. The experimental group completed assessments at three time points: Time 1 (baseline/immediately before intervention); Time 2 (post-test 1 immediately after the intervention), and Time 3 (post-test 2 three months after intervention). The control group completed measures at two time points: Time 2 (baseline) and Time 3 (3 months later). Head nurses of participants completed measures at time points that matched the respective group that the participant was assigned to, either experimental (Time 1/baseline, Time 2/post-test 1, Time 3/post-test 2) or control (Time 2/baseline, Time 3/3 months later). The research assistant distributed surveys to the following groups at the following times: experimental group (Time 3); control group (Time 2 and Time 3), and head nurses (all time points). Research assistants facilitated obtaining informed consent from the control group and distributing study incentives.

For the experimental group, relevant study information was explained to participants before the intervention. After obtaining informed consent, the background information, SPS, SCPS-R, and self-evaluated SCCS, was distributed (via paper and pencil) and collected before the intervention (Table 1 & Figure 1). After the morning session of intervention, but before the OSCE, the experimental group completed the SPS, SCPS-R, self-evaluated SCCS, and individual reflection logs (EG Time 2 measures). Experimental groups completed measures again at the 3-months post-intervention period (Time 3); the collection of surveys was facilitated by the research assistant. Upon completion of surveys at Time 3, the EG received an additional 200-New Taiwan Dollars as a token of appreciation for their participation in the study.

Control group participants completed paper and pencil copies of the background information and self-evaluated SCCS, SPS, SCPS-R at Time 2 (baseline for CG). Instruments were collected in a secure box located on the respective units. Within a week’s time, the control group participants were awarded a 200-New Taiwan dollar gift card as a token of appreciation for their participation by the research assistant. This process was repeated at Time 3 (3-month time point), without the background information survey, and was facilitated by the research assistant, who also distributed an additional 200-New Taiwan dollar gift cards to show appreciation for the CG’s completion of surveys at Time 3.

The head nurses of the study participants received e-mails to explain the relevant information of this study and to obtain signed informed consent. The HN-evaluated SCCS was filled out for experimental group participants (Time 2, Time 3) and for control group participants (Time 2 and Time 3). Head nurses were given a 200-New Taiwan dollar gift card by the research assistant as a token of appreciation for their participation for each time period they completed surveys. 

### 2.6. Data Analysis

All data were analyzed using SPSS 22.0 Windows software (IBM Corporation, Armonk, NY, USA). The assumptions for normality, outliers, and multicollinearity were checked and two outliers were deleted using Winsorizing for parametric analyses. The descriptive statistics reported frequency, percentage, mean ± standard deviation (SD), and range. Inferential statistics included linear regression, unpaired *t* test, chi-squared and Fisher’s exact test, repeated measures ANOVA, and paired *t* test. Significance level was set as *p* < 0.05. Content analysis was applied to analyze reflection logs.

### 2.7. Ethical Considerations

Ethical approval was obtained from the Institutional Review Board (No. 20171612b0c501) after reviewing and approving the study protocol. All participants completed signed informed consent. Participants all volunteered to participate in this study and were able to withdraw from this study at any time. Questionnaires were collected anonymously.

## 3. Results

### 3.1. Background Information

The nurse participants’ ages ranged from 22–56 years for the entire sample (mean age 31.69 ± 7.72) and 98.6% of nurses were female (Table 3). Most nurses were single (73.2%) and over one-third of the nurses graduated from two-year colleges (43.5%). Greater than half reported having a religion (58.7%). More than one-third of the nurses’ clinical ladder positions were ≥N4 (36.2%). The only significant difference between groups found in background information was in interest in spirituality and spiritual care (*p* < 0.001). Thus, interest in spirituality and spiritual care and baseline scores of each outcome were treated as covariates for analyzing between-subject effects.

### 3.2. SPS, SCPS-R, and Self- and HN-Evaluated SCCS at Baseline

At baseline (Table 4), the mean SPS score (beliefs and spiritual behavior engagement) of the experimental group was slightly higher than that of the control group (37.60 ± 10.25 vs. 36.51 ± 6.85, respectively); the mean SCPS-R (spiritual care perspectives) score of the experimental group was slightly lower than that of the control group (38.45 ± 2.78 vs. 39.28 ± 3.13, respectively), and the mean self-evaluated SCCS (spiritual care competence) score of the experimental group was lower than that of the control group (86.15 ± 15.33 vs. 89.15 ± 13.58, respectively). No significant differences were found between the two groups. However, the mean HN-evaluated SCCS score of the experimental group (102.39 ± 15.76) was significantly higher (t_(133)_ = −3.04, *p* = 0.003) than that of the control group (94.23 ± 14.74, respectively). Therefore, the baseline of mean HN-evaluated SCCS scores was treated as a covariate for examining the between-subject effects at Time 3.

### 3.3. Between-Subject Effects of the SPS, SCPS-R, and Self- and HN-Evaluated SCCS

Table A2 shows linear regression analysis of the mean SPS score (spiritual beliefs and spiritual behavior engagement) at Time 3 (3 months) of the experimental group, which was significantly higher than that of the control group at Time 3 (3 months) (42.45 ± 9.73 vs. 37.42 ± 6.32, respectively; *b* = 2.86, *p* = 0.012) after adjusting for SPS baseline scores and interest in spirituality and spiritual care. The mean SCPS-R score (spiritual care perspectives) at Time 3 of the experimental group was significantly lower (indicating more positive spiritual care perspectives) than that of the control group (37.17 ± 2.48 vs. 39.06 ± 2.90, respectively; *b* = −0.98, *p* = 0.015) after adjusting for covariates. The experimental group had a significantly higher mean self-evaluated SCCS score (spiritual care competence) at Time 3 than the control group after adjusting for covariates (98.85 ± 16.14 vs. 91.21 ± 13.87, respectively; b = 7.05, *p* = 0.002). At Time 3, experimental and control group HN-evaluated SCCS scores were not significantly different after adjusting for covariates (102.39 ± 15.76 vs. 94.23 ± 14.74, respectively; *b* = −0.98, *p* = 0.695).

### 3.4. Within-Subject Effects on the SPS, SCPS-R, and Self- and HN-Evaluated SCCS

The RM ANOVA (Table A3) showed significant differences between the overall mean SPS (beliefs and engagement in spiritual behaviors) (*F*_(1_._57,81_._48)_ = 17.12; *p* < 0.001, η^2^ = 0.248), SCPS-R (spiritual care perspectives) (*F*_(2,104)_ = 29.93; *p* < 0.001, η^2^ = 0.365), and self-evaluated SCCS scores (spiritual care competence) (*F*_(2,104)_ = 36.68; *p* < 0.001, η^2^ = 0.414) of the experimental group across the three time points. Post hoc analysis showed that the mean SPS scores at Time 2 and Time 3 were significantly greater than the mean baseline (Time 1) score, respectively (*p* < 0.001). The mean SCPS-R scores at Time 2 and Time 3 were significantly lower than the mean baseline (Time 1) score, respectively (*p* < 0.001), while the mean SCPS-R score at Time 2 was significantly lower than that at Time 3 (*p* = 0.001). Lower scores indicated more positive spiritual care perspectives. The mean self-evaluated SCCS scores at Time 2 and Time 3 were significantly higher than the baseline (Time 1) score, respectively (*p* < 0.001). However, the paired *t* test showed no significant differences in the mean HN-evaluated SCCS (spiritual care competence) scores at Time 1 and Time 3 (Table A4).

In Table A4, the paired *t* test showed no significant differences in the mean SPS (36.51 vs. 37.42, t_(84)_ = −1.44; *p* = 0.154), SCPS-R (39.28 vs. 39.06, t_(84)_ = 0.79; *p* = 0.434), and self-evaluated SCCS scores (89.15 vs. 91.21, t_(84)_ = −1.62; *p* = 0.109) at the baseline and second posttest of the control group. The only significant difference for the CG was in mean HN-evaluated SCCS between the baseline and second posttest (93.78 vs. 97.17, t_(82)_ = −2.14; *p* = 0.035).

### 3.5. Course Satisfaction, OSCE, and SPFS of the Experimental Group

For the experimental group, mean course satisfaction scores for nine items ranged from 18–45 with a grand mean of 38.87 ± 5.38 and an average of 4.34 divided by 9 items. Most nurses (91.8%) were satisfied or very satisfied with the educational program. The mean overall satisfaction score was 4.34 ± 0.62, with 92.5% of nurses satisfied or very satisfied with this program. The items with the highest mean satisfaction scores were: “Course content covers the importance of clinical spiritual care” (4.47 ± 0.61), “Course content is practical for clinical spiritual care” (4.43 ± 0.64), and “The real scenario helps me to learn clinical spiritual care competence” (4.40 ± 0.69). The items with the lowest mean satisfaction scores were: “Course content increases my confidence in patients’ spiritual care” (4.25 ± 0.68), “I can apply the spiritual care skills of the course content to the clinical spiritual care process in the future” (4.25 ± 0.68), “Teaching methods help me to provide patients’ spiritual care process in the future” (4.25 ± 0.68), “The application of teaching strategies helps me to understand the course content” (4.25 ± 0.71), and “The course content increases my clinical spiritual care competence” (4.26 ± 0.68). Qualitative feedback on this course that described areas of participant satisfaction included clinical cases sharing (45.3%, *n* = 24), spiritual care situation simulation video (26.4%, *n* = 14), awareness of the definition and scope of spirituality (7.5%, *n* = 4), and life review skills (7.5%, *n* = 4). 

The mean OSCE scores of the EG for the nine items ranged from 1–17 with a grand mean of 11.34 ± 3.58 (Table A5). The majority of nurses had completed all nine items as intended (44.3%), this was followed by partially completed nine items (37.5%). The highest mean OSCE scores were for the items: “Demonstrate eye contact and listening behavior” (1.81 ± 0.40), “Encourage patients to express their thoughts and feelings about having cancer” (1.70 ± 0.50), and “Can soothe the patient’s emotions in a timely manner” (1.57 ± 0.54). The lowest mean OSCE scores were: “Can relate to loss as a normal reaction” (0.47 ± 0.70), “Can apply body language to interact with patients” (0.68 ± 0.80), and “Can assess relevant data of the spiritual aspect” (0.85 ± 0.50). The mean global performance scores were 3.40 ± 0.91 and global performance was ranked by “passed” (43.4%), “good” (35.8%), “passing edge” (11.3%), “excellent” (7.5%), and “failed” (1.9%).

The mean standardized patient feedback scale (SPFS) scores of thirteen items ranged from 15–76 with a grand mean of 62.23 ± 10.33. The 13-item SPFS included partially agree 64.5%, while 26.1%was strongly agree. The highest mean SPFS scores were for: “Examinee willing to take time to speak with me” (5.06 ± 0.93), “I can understand the semantics of the examinee (5.04 ± 1.02)”, and “Examinee performs well in nursing professional standards (4.94 ± 1.05). The lowest mean SPFS scores were for: “Examinees address patients by their names and honorific titles to confirm the patients’ identity (4.30 ± 1.81)”, “Examinees can introduce themselves (4.38 ± 1.92)”, and “Examinees have stated the purpose of the interview (4.62 ± 0.97).

### 3.6. Reflection Logs 

Content analysis revealed (Figure A1) the common subthemes of the two groups “what I had seen, heard, and touched from clinical spiritual care practice content”; “what had brought my feelings, thoughts, learning, and meanings from clinical spiritual care process”, and “what had changed my view on things, the world, self, beliefs, and life’s meaning and what I can do” at the end of the intervention (Time 2). What can be done includes spiritual assessment, spiritual care skills, reflection on spiritual care experience during work, improving and refining spiritual care competence, knowledge and competence and insufficient confidence and difficulty of providing clinical spiritual care, the function and importance of religion, function of spiritual care, and absence of reflection. However, the experimental group had higher mean percent across “what, so what, and now what” than the control group on spiritual assessment (20.1% vs. 0.8%), spiritual care skills (37.7% vs. 20.4%), reflection on spiritual care experience during work (30.8% vs. 5.5%), spiritual care competence needs to be improved and refined (28.9% vs. 6.7%), and knowledge and competence and confidence insufficiency and difficulty of clinical spiritual care (28.3% vs. 15.3%). On the contrary, the control group had higher mean percent across “what, so what, and now what” than the experimental group on function and importance of religion (0.9% vs. 12.4%), function of spiritual care (8.2% vs. 13.3%), and absence of reflection (1.3% vs. 13.3%). 

After 3 months (Time 3), the experimental group still had higher mean percent across “what, so what, and now what” than the control group on spiritual assessment (17.0% vs. 2.4%), spiritual care skills (36.5% vs. 9.8%), reflection on spiritual care experience during work (22.6% vs. 4.3%), spiritual care competence needs to be improved and refined (20.1% vs. 2.7%), and knowledge, competence, and confidence insufficiency and difficulty of providing clinical spiritual care (16.0% vs. 14.1%). On the contrary, at Time 3, the control group had higher mean percent across “what, so what, and now what” than the experimental group on function and importance of religion (4.7% vs. 13.8%), function of spiritual care (9.4% vs. 12.2%), and absence of reflection (9.4% vs. 18.0%). 

In summary, this intervention enhanced nurses’ spiritual perspective, self-perceptions and HN-evaluated spiritual care competence, and improved spiritual care perspective. Overall, 91.8% of the nurses that received the intervention were satisfied or very satisfied with this educational program. The global performance mean score of OSCE was 3.40 ± 0.91(range 1–5) and 86.8% (*n* = 46) of EG participants passed the OSCE. The grand mean of standardized patient feedback scores was 62.23 ± 10.33 (range 15–76). In the experimental group reflection logs, the most common subthemes were reflections on spiritual care skills; reflection on spiritual care experience during work; knowledge, competence, and confidence insufficiency and difficulty of clinical spiritual care, and spiritual assessment.

## 4. Discussion

This study employed scenario simulation and OSCE with standardized patient feedback to explore the effects of a scenario-based spiritual care course on spiritual care competence in clinical nurses. The study supported the use of these teaching methods to improve spiritual care outcomes. The between-subject effects showed that the experimental group had significantly higher scores related to spiritual beliefs and spiritual behavior engagement (SPS) and self-evaluated spiritual care competence (SCCS) and more positive spiritual care perspectives (SCPS-R) than the control group at three months after adjusting for covariates. These results emphasized that the scenario-based spiritual care educational program enhances nurses’ spiritual perspective, positive spiritual care perspective, and spiritual care competence. However, the two groups had no significant differences in HN-evaluated SCCS (spiritual care competence) at Time 3 after adjusting for covariates. The discrepancies between self- and HN-evaluated SCCS may have occurred because head nurses and nurses employed subjective and intuitive assessment of their spiritual care competence through determining and weighting the diverse information used for spiritual care assessment. Future studies might consider subjective and objective measures of spiritual care competence for comparing between-subjects effect. Objective measures may include direct observation of real patients for bedside evaluation on both groups [57]. The authors were unable to find published studies that explored the between-subject effects of the SPS and SCPS-R to use for comparison to the present study’s results. The finding of this study’s self-evaluated SCCS scores were similar to those of Hu et al. [30], which used a spiritual care education curriculum that included lectures, case sharing, group discussion, individual psychological counselling with centralized training, and organized activities over one day (eight hours) lasting five days with two sessions every six months for one year among cancer nurses. Nonetheless, the findings of Hu et al. [30] had statistical issues without adjusting differences in total scores of SCCS at the baseline. 

In this study, the within-subject effects showed that the experimental group had gradually increasing spiritual beliefs and spiritual behavior engagement scores and self-evaluated spiritual care competence scores that were significant from immediately after the educational program to three-months post intervention. In addition, improvements in positive spiritual care perspectives from immediately after to three months post program were also seen. This may be attributed to the effects of the scenario-based spiritual care educational program. Although the HN-evaluated spiritual care competence scores had increased for the experimental group at three months, no significant differences were reached between Time 2 and Time 3. However, no significant differences were seen in any of the measures (SPS, SCPS-R, SCCS) for the control group between baseline (Time 2) and three months thereafter (Time 3). Nevertheless, the HN-evaluated spiritual care competence scores of the control group had increased at Time 3 and reached significant difference. Similar to O’Shea and colleagues [9], who saw improved spiritual care perspectives (SCPS-R) in pediatric nurses after an educational intervention, the present study saw marked and significant improvements of SCPS-R scores in the EG at Time 2 and Time 3 compared to the baseline. This finding of the within-subject effect on the experimental group was also demonstrated by Hu et al. [31] and Petersen [34,35], but the control group results of Hu et al. [31] were different from the present study. The control group findings of Hu et al. [31] found significantly higher total spiritual care competence (SCCS) scores but the magnitude of difference was less than that of the experimental group, which may have been attributed to participation in the centralized study of the hospital for only one session (2 h) monthly during the study period. Likewise, Petersen and Petersen et al. [34,35] found that, after an educational intervention, spiritual care competence (SCCS) total scores were significantly different from baseline, immediately after the program, and three months after the program. In addition, Petersen’s post hoc analysis showed that scores immediately after the program and three months after the program were significantly higher than the baseline spiritual care competence scores. Unlike the present study, Petersen and Petersen et al. [34,36] also found scores at three months after the program were significantly higher than immediately after the program, whereas the present study did not see significant differences between these two time points.

Experimental group nurses were satisfied to very satisfied or satisfied (91.8%) with the scenario-based simulation program. The top three areas of nurses’ satisfaction were related to the course content, practice implications for clinical spiritual care, and relationship of the authentic simulation to learning spiritual care competence. After participating in the program, experimental group nurses’ lowest scoring items on the program evaluation, which still mostly indicated positive feelings of “agree” to “strongly agree” score range, related to confidence in spiritual care skills, application of course skills to clinical spiritual care practice, teaching methods, teaching strategies, and perceived increase in competence related to knowledge from course. For some nurses, the combination of lecture, video, simulation, and OSCE with SP may not have been a preferred learning format, or they did not believe this blended model course provided material that improved application, confidence, and competence related to spiritual care. For some participants, it was still difficult to apply spiritual care skills to their own clinical situation based on a four-hour course. However, OSCE with SP gave nurses the opportunity to practice and to apply the morning spiritual care knowledge attained to skills in the afternoon. Future studies may cultivate embedded teachers at the point of care for spiritual assessment instruction, writing case reports, conducting bedside teaching on providing spiritual care, and evaluating real-time spiritual care competence using OSCE rating scales and providing feedback from real patients and embedded teachers [58].

The OSCE showed nurses’ spiritual assessment performance ranged from 1 to 17 with a nine-item checklist. Most nurses completed the nine items; however, others were only able to demonstrate partial achievement of satisfactory skills. Nurses had better performance with eye contact and demonstrating listening behaviors, encouraging patients to express their thoughts and feelings about being diagnosed with cancer, and helping to address the patients’ emotions on a timely basis. Nurses were weaker in performance relating to loss as a normal reaction, applying body language to interact with patients, and assessing data related to the dimension of spirituality. Fewer nurses used body language to interact with the patient, which may have been the result of seeing the patient for the first time and possibly because the OSCE was conducted during the COVID-19 pandemic. Upon the first patient meeting and with new clients, Taiwanese nurses do not typically interact with patients using body language, which may be related to Chinese culture [59]. In addition, nurses had a hard time completely grasping the dimension of spirituality after only four hours of training; it was difficult for some to apply spiritual assessment and skills to the scenario of the OSCE. The majority of examinees passed the global OSCE performance (43.4%) or demonstrated good OSCE performance (35.8%). However, there were still a number of participants that did not perform as expected, and these performances reflect that some nurses were not familiar with spiritual assessment for a young woman with gynecological cancer. An improvement in spiritual care competency needs to constantly practiced at the workplace and one’s own lived experience with spiritual and spiritual care perspective needs to be reflected on, as does experiences with spiritual care. Like many areas of clinical practice, spiritual care skills require nurturing, continual reflection, and ongoing attention to areas related to spiritual care competence and confidence.

From the perspectives of the SPs, the majority of examinee scores (64.5%) ranked in the “partially agree” category, whereas only 26.1% fell within the “strongly agree” category related to OSCE performance. The SPs felt the nurses provided time talking with them as a patient, that they understood the nurses’ actions, and that the nurses were acting as professionals. Areas where SPs indicated the experimental group was lacking were related to introducing self, using patient full name and title, and explaining why the SP was being interviewed. These SP feedback comments may reflect that nurses were used to demonstrating certain behaviors in the workplace and perhaps they did not consider fundamental elements of introduction and describing purpose of action as relevant in a simulated experience. Although the OSCE checklist showed good content validity and construct validity, future studies may conduct a Delphi survey on items for evaluating spiritual assessment competence or spiritual care competence by different nursing clinical ladder and applying entrusted professional activities (EPAs) to bedside teaching and competence evaluation.

From the reflection logs, the experimental group had a higher mean percent of subthemes on the reflections across “what, so what, and now what” than the control group on spiritual assessment, spiritual care skills, reflection on spiritual care experience during work, spiritual care competence needs to be improved and refined, and knowledge and competence and confidence insufficiency and difficulty of clinical spiritual care. The control group had higher mean percent of subthemes on the reflections across “what, so what, and now what” than the experimental group on function and importance of religion, function of spiritual care, and no reflection. These were likely due to the effects of the scenario-based spiritual care course and previous clinical spiritual care experience in the experimental group. Most nurses in the control group did not fully understand spirituality, they usually viewed religion as equal to spirituality, and fewer control group nurses reflected on prior spiritual care experiences. 

After three months, the experimental group still had a higher mean percent of subthemes across “what, so what, and now what” than the control group on spiritual assessment, spiritual care skills, reflection on spiritual care experience during work, spiritual care competence needs to be improved and refined, and knowledge, competence, and confidence insufficiency and difficulty of clinical spiritual care, whereas the control group had higher mean percent of subthemes across “what, so what, and now what” than the experimental group on function and importance of religion, function of spiritual care, and no reflection. These results demonstrated the influence of the scenario-based spiritual care program and previous clinical spiritual care experience in the experimental group. Most nurses in the experimental group tried to apply the knowledge and skills of this course to clinical spiritual care for their patients and families and reflected on their spiritual care experiences. However, after three months, most nurses in the control group still did not completely understand spiritual care and even fewer nurses reflected on prior spiritual care experiences. 

Both groups depicted spiritual growth and joy more at Time 3 than at Time 2 or baseline. In addition, both groups indicated the need to improve and increase spiritual care competency and recognized having insufficient spiritual care knowledge and skills and acknowledged difficulty in clinical application. Although both groups had various reflection subthemes, both groups had positive and negative subthemes. Nurses had their own spiritual or spiritual care perspectives, appreciated spiritual or spiritual care function and importance, experienced spiritual growth and joy, and identified their own weakness and strengths on spiritual care. Therefore, ward or nursing department-based in-service education is needed to adopt teaching strategies of showing “how” (performance) and “does” (action), presented as part of Miller’s pyramid of professional competence, a framework for assessing skills, in order to help clinical nurses break through their dilemma of providing spiritual care [60].

### Limitations

This study had several limitations. First, this quasi-experimental study without randomization may compromise internal validity; however, matching was used to overcome this issue. Second, the experimental group had fewer nurses than the control group, despite our best efforts to recruit participants and study period postponement during the COVID-19 pandemic. Third, this study recruited participants from three branches of a large medical foundation, which may limit generalization of results to other populations. Selection bias cannot be ruled out, although covariates were controlled by statistical analysis. Fourth, a change in spiritual care behavior of the experimental group after the educational program in the unit may have affected other unit nurses in the control group. However, to develop spiritual care competence, clinical nurses must learn and reflect constantly on the aspects of spiritual care while providing clinical care for patients. Nurses may cultivate spiritual care competence through educational programs that represent the most fundamental learning strategies [13,29].

## 5. Conclusions

The scenario-based spiritual care course effectively enhanced clinical nurses’ spiritual perspectives and spiritual care competence. Future studies and in-service education may support spiritual care competence by cultivating clinical nurse preceptors with better spiritual care pedagogy through the use of EPAs for bedside teaching during nurses’ postgraduate years. With increased spiritual care competence, nurses may deliver spiritual care, achieve spiritual growth and joy, and provide holistic patient care.

## Figures and Tables

**Figure 1 healthcare-11-00036-f001:**
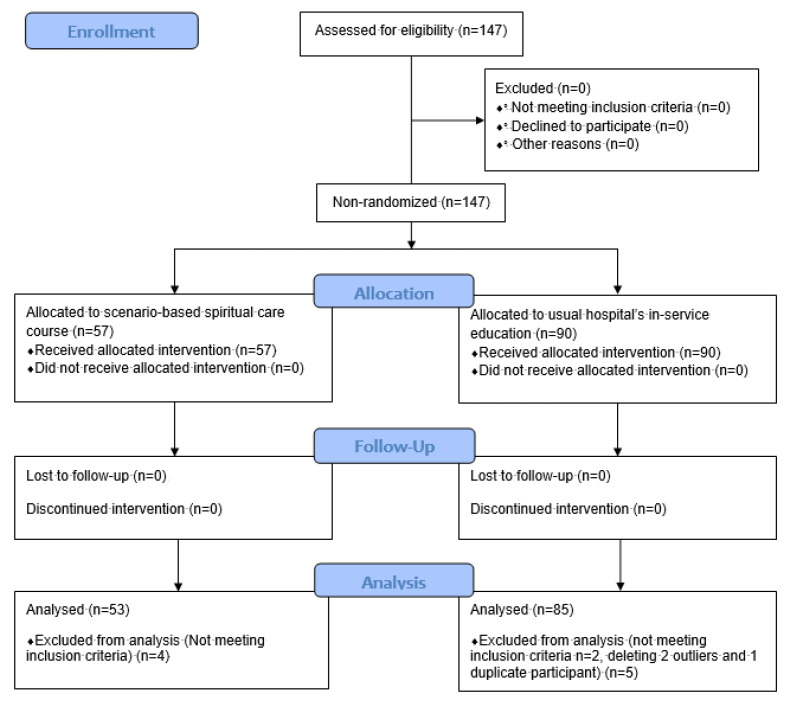
Flow diagram of the study.

**Table 1 healthcare-11-00036-t001:** Study Design and Collection Points of Measurements.

NR	O_1_	X	O_2_	O_3_
NR			O_1_	O_2_
Measures	Time 1		Time 2	Time 3
1. Background information	√(EG)		√(CG)	
2. Self-evaluated SCCS (PO)	√(EG)		√(EG & CG)	√(EG & CG)
3. HN-evaluated SCCS (PO)			√(EG & CG)	√(EG & CG)
4. SPS (PO)	√(EG)		√(EG & CG)	√(EG & CG)
5. SCPS-R (PO)	√(EG)		√(EG & CG)	√(EG & CG)
6. Reflection log (PO)			√(EG)	√(EG & CG)
7. Course Satisfaction (SO)			√(EG)	
8. SPFS (SO)			√(EG)	
9. OSCE Checklist (SO)			√(EG)	

Note. NR, non-randomization; O_1_, baseline measure (EG); O_2_, measure just after intervention (EG + HN) and baseline (CG + HN); O_3_, measure after 3 months(EG, CG, HN); HN, Head Nurse of participant in experimental or control group; SCCS, Spiritual Care Competence Scale; SPS, Spiritual Perspective Scale; SCPS-R, Spiritual Care Perspective-Revision; SPFS, Standardized Patient Feedback Scale; OSCE, Objective Structure Clinical Examination; PO, primary outcome; SO, secondary outcome; EG, experimental group; CG, control group. Time 1 is pre-intervention for EG; Time 2 is post-intervention for EG and baseline for CG; Time 3 is 3 months post-intervention for EG and 3 months after baseline for CG.

**Table 2 healthcare-11-00036-t002:** The Spiritual Care Course Objectives, Teaching Content and Strategies, and Schedules.

	Content
Learning objectives	1. Differentiate the definitions of spirituality, religion, and spiritual care. 2. Understand the impact of illness. 3. Understand clues of patient or family conversation about spirituality. 4. Perform spiritual care assessment. 5. Clarify spiritual distress and well-being. 6. Execute the nursing process of spiritual care. 7. Write individual spiritual reflection logs. 8. Understand skills of looking for life’s meaning and purpose.
Content and related teaching strategies	Morning Session 1. Lecture with PowerPoint, handout, and shared cases. Spiritual impact of illness; definitions of spirituality, religion, and spiritual care Spiritual clues of patients and families Spiritual needs assessment; spiritual care nursing process; spiritual distress and wellbeing; spiritual care methods and skills Searching life’s meaning and purpose; participant spiritual reflection 2. Scenario-based spiritual care video and debrief. Case: 30-year-old female with palpitations was admitted for cardiac catheterization. After catheter ablation treatment, complications occurred, and she developed a 3rd degree of A-V block (heartbeat 40 times per minute with fatigue). In this scenario, the patient experienced shortness of breath, dyspnea on exertion, and thought about the meaning of her life. A negative and positive nursing spiritual care performance for the case was presented on the video. Debrief: Five questions regarding this scenario for discussion. Afternoon Session 1. OSCE Case: the scenario for spiritual assessment was a 19-year-old girl newly diagnosed with cervical cancer. She was admitted for hysteroscopic examination and chemotherapy. SP provided feedback after the examination
Course Offerings and Participants	1. 17 August 2019–11 January 2020 (7 times): held 5 times on 17 August (*n* = 4); 28 September (*n* = 5); 2 November (*n* = 5); 23 November 2019 (*n* = 8); & 11 January 2020 (*n* = 5) 2. 22 February 2020–13 June 2020 (5 times): held 2 times on 18 April (*n* = 3); & 16 May (*n* = 1) 3. 29 August 2020–28 November 2020 (3 times): held 3 times on 29 August (*n* = 11); 31 October (*n* = 9); & 28 November (*n* = 6)
Course Schedule	08:00–08:20 (20 min) Check in and completion of baseline measures 08:21–10:10 (110 min) Lecture of spiritual care 10:11–10:20 (10 min) Rest/Break 10:21–11:20 (60 min) Scenario-based spiritual care video watching and debriefing 11:21–11:30 (10 min) Rest/Break 11:31–12:00 (30 min) Completion of 1st posttest measures 12:01–13:00 (60 min) Rest and lunch 13:01–16:00 OSCE (15 min examination and 5 min SP feedback) 16:01 Check out and distribute gift card (NT 500 dollars)

**Table 3 healthcare-11-00036-t003:** Homogeneity test of sociodemographic variables of clinical nurses by group at baseline (*n* = 138).

		Control	Experimental		
		(*n* = 85)	(*n* = 53)	t/χ^2^ Test	*p* Value
Variable	Category	*n* (%)	*n* (%)		
Age in years ^a^				t_(136)_ = −0.62	0.535
Range		22–55	23–56		
Mean (SD)		32.01 (7.45)	31.17 (8.19)		
Gender ^b^	Male	2 (2.4)	0 (0.0)	NA	0.523
	Female	83 (97.6)	53 (100.0)		
Marital status ^c^	Single	61 (71.8)	40 (75.5)	χ^2^_(1)_ = 0.23	0.633
	Married	24 (28.2)	13 (24.5)		
Education ^c^	2-year/5-year diploma	4 (4.7)	7 (13.2)	χ^2^_(3)_ = 4.00	0.261
	2-year college	39 (45.9)	21 (39.6)		
	4-year college	8 (9.4)	7 (13.2)		
	≥University/master’s programme	34 (40.0)	18 (34.0)		
Religion ^b^	None identified	38 (44.7)	19 (35.8)	8.18	0.135
	Buddhist	6 (7.1)	8 (15.1)		
	Christian/Catholic	4 (4.7)	7 (13.2)		
	Taoist	23 (27.1)	8 (15.1)		
	I-Kuan Tao	2 (2.4)	2 (3.8)		
	Folk beliefs	12 (14.1)	9 (17.0)		
Working years ^a^				t_(136)_ = −0.65	0.519
Range		0.1–35.8	0.3–34.1		
Mean (SD)		9.42 (7.62)	8.55 (7.85)		
Nursing clinical ladder ^c^	N0	7 (8.2)	3 (5.7)	χ^2^_(4)_ = 4.94	0.294
	N1	7 (8.2)	10 (18.9)		
	N2	21 (24.7)	16 (30.2)		
	N3	17 (20.2)	7 (13.2)		
	≥N4	33 (38.8)	17 (32.1)		
Physical health status ^c^	Very poor/poor	9 (10.6)	1 (1.9)	χ^2^_(2)_ = 4.12	0.128
	Common	58 (68.2)	37 (69.8)		
	Good/very good	18 (21.2)	15 (28.3)		
Interest in spirituality and spiritual care ^a^				t_(136)_ = −5.40	**<0.001**
Range		2–5	3–5		
Mean (SD)		3.32 (0.60)	3.92 (0.70)		

Note ^a^ unpaired *t* test; ^b^ Fisher’s exact test; ^c^ chi-square test; NA, Not Applicable. The bold *p* value isstatistically significant. Nursing clinical ladder: N0: <1-year clinical experience. N1: 1-year clinical experience, completion of N1 clinical professional competence training, pass N1 review, and can perform basic patient/client care. N2: 2-years clinical experience, completion of N2 clinical professional competence training, pass N2 review, and can perform critical patient/client care. N3: 3-years clinical experience, completion of N3 clinical professional competence training, pass N3 review, can perform critical patient/client holistic care, and have the competence of teaching and assist the working unit in improving quality. N4: 4-years clinical experience, completion of N4 clinical professional competence training, pass N4 review, can perform critical patient/client holistic care, and have the competence of teaching, participating in administration and perform the working unit’s quality improvement [57].

**Table 4 healthcare-11-00036-t004:** Homogeneity test of outcomes at baseline (*n* = 138).

	Control	Experimental	t/Fisher’s	
	(*n* = 85)	(*n* = 53)	Exact Test	*p* Value
Variable	*n* (%)	*n* (%)		
SPS ^a^			t_(81_._1)_ = 0.69	0.492
Range	20–57	19–59		
Mean (SD)	36.51 (6.85)	37.60 (10.25)		
SCPS-R ^a^			t_(136)_ = 1.58	0.116
Range	32–48	33–44		
Mean (SD)	39.28 (3.13)	38.45 (2.78)		
Self-evaluated SCCS ^a^			t_(136)_ = 1.20	0.232
Range	55–125	52–109		
Mean (SD)	89.15 (13.58)	86.15 (15.33)		
Self-evaluated SCCS ^b^			2.09	0.381
Low competence (<64)	2 (2.4)	4 (7.5)		
Moderate competence (64–98)	60 (70.6)	36 (67.9)		
High competence (99–135)	23 (27.1)	13 (24.5)		
HN-evaluated SCCS ^a^			t_(133)_ = −3.04	**0.003**
Range	63–131	66–129		
Mean (SD)	94.23 (14.74)	102.39 (15.76)		
HN-evaluated SCCS ^b^			4.48	0.072
Low competence (<64)	1 (1.2)	0 (0.0)		
Moderate competence (64–98)	52 (61.9)	23 (45.1)		
High competence (99–135)	31 (36.9)	28 (54.9)		

Note. ^a^ unpaired *t* test; ^b^ Fisher exact test; SPS, Spiritual Perspective Scale; SCPS-R, Spiritual Care Perspective-Revision; SCCS, Spiritual Care Competence Scale; HN, Head Nurse; Higher SPS or SCCS scores indicate a greater spiritual perspective or spiritual care competence; Lower SCPS-R scores indicate a positive spiritual care perspective. Baseline was Time 1 for EG and Time 2 for CG. The bold *p* value is statistically significant.

## Data Availability

The data are not publicly available due to ethical restrictions.

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
