# Peer review of "The Effects of a Scenario-Based Spiritual Care Course on Spiritual Care Competence among Clinical Nurses: A Quasi-Experimental Study"

_healthcare, 2022, doi:10.3390/healthcare11010036_

Round 1
Reviewer 1 Report
The Effects of a Scenario-Based Spiritual Care Course on Spiritual Care Competence among Clinical Nurses: A Quasi-Experimental Study
Reviewer comments
General
The authors are commended for their work on the topic of spiritual care competence. An area that needs to be addressed. Thank you for your time and effort.
The reviewer understands that the design cannot be changed at this point of time but does question why the pretest was not administered at the same time to the control and experiential groups. A change in behavior of the experimental group following the education on the unit may have had an influence on co-workers in the control group. This can be addressed in the limitations of the study.
Consider changing the title of the “pretest” for the control group to clarify that it was not prior to the intervention rather reflects that it was 1st time administered and correlated with the 1st post test for the experimental group.
The reviewer also has a question on the approach used to train the head nurses in the completion of the Spiritual Care Competence Scale (SCCS). Could their lack of understanding of the rating scale potentially influenced the results?
Tables are difficult to read with lack of separation of the content for different categories.
Specific comments
-
Brief statement on matching process is needed as well as clarification of what the 1-3 ratio means.
control group was matched for similar units, ages, working experience, and 88 nursing clinical ladder with the experimental group (1:1-3 ratio).
-
Table 1 is confusing. Be clear when the measures were administered to the control group and to the experimental group. Table 1 line 108
Table 1. Study design and measurements
-
Clarify “longer and shorter periods of time. Line 116 plus
longer periods of time were included. Nurses and nursing managers who worked in the outpatient department, health checkups, emergency department, operating room, hemodialysis and peritoneal dialysis units, and chem-118 otherapy room providing task oriented and shorter time care were excluded
-
Clarify - I believe the authors mean to indicate that there was no previously reported effect size in the literature Line 119
No previous effect size of between-subjects effects was used to estimate the sample size.
-
How many times was the program offered? Recommend that the term “session” be replaced by program as session indicates to the reviewer that participants attended numerous times. Line 129
The scenario-based spiritual care course was about four hours each session, including a pretest, a lecture using
-
Clarify what the 1-3 means in the statements below Line 143 plus I believe that the authors are indicating that there were 3 examiners and there was a total of 3 standardized patients
Examiners (n = 1-3) watched good and poor versions of a video two days before the OSCE examination for assessing inter-rater reliability. Standardized patients (n = 1-3) were qualified and trained before the OSCE examination.
-
Was it seven or six?? Line 159
Then, seven or six experts on spiritual care practice, hospice and palliative care, and Buddhist chaplain in the clinical
-
The amount of dollars distributed to the participants would be extreme for the organization I have been affiliated with. Are the amounts correct? Lines 172 to 181
500-dollar gift card in the afternoon. Experimental group participants helped to find nurses from the same units or division, with same age range, nursing clinical ladder, working years, and education who understood this study and were willing to fill out the spiritual care-related scales as a matched control group. After completing the scales and collecting them in a cardboard box within a week’s time, the control group participants were awarded a 200-dollar gift card. The head nurses of the experimental group received e-mails to explain the relevant information of this study and to obtain signed informed consent. The HN-evaluated SCCS was filled out and head nurses were given a 200-dollar gift card. Then, the SPS, SCPS-R, self- and HN-evaluated SCCS, and reflection log were collected from both groups three months after the in-181 intervention and were given a 200-dollar gift card.
-
Sentence edits - repetitive use of “more” in the section that follows. Unsure what the N4 means at the end of the section that follows- Line 202-204
Most nurses were single (73.2%) and more 202 than one-third of the nurses graduated from two-year colleges (43.5%). More than half 203 reported having a religion (58.7%). More than one-third of the nurses’ clinical ladder po-204 sitions were N4 (36.2%).
-
Table 2 - recommend that the table be formatted so that the variables stand out. Currently the variables are in the same list as the variable options.
-
In table 2 one of the options for Religion is No - consider changing to “None identified”
Religion No
-
In table 2 the term “Nursing Clinical Ladder” is used as a variable header. What do the options ries mean?
Nursing clinical ladder
-
“Had no” - change wording to “was not” Line 234
Additionally, the experimental group had no significantly higher mean HN-evaluated SCCS scores at the 2nd posttest than the control group after adjusting for covariates (102.3915.76 vs. 94.2314.74, respectively; b = -0.98, 236 p = .695).
-
Results are presented logically. One thing that would be helpful is to note the meaning of the scale results as not all of the scales have the same direction (higher rating reflects higher spiritual competence, etc). It is on the measures table but telling the reader this in the results section would be helpful.
Author Response
Point 1: General
The authors are commended for their work on the topic of spiritual care competence. An area that needs to be addressed. Thank you for your time and effort.
Response 1.1: Thanks for the comments. A paragraph of spiritual care competence has added to the introduction (p.3 line 106-123)
The reviewer understands that the design cannot be changed at this point of time but does question why the pretest was not administered at the same time to the control and experiential groups. A change in behavior of the experimental group following the education on the unit may have had an influence on co-workers in the control group. This can be addressed in the limitations of the study.
Response 1.2: This study recruited experimental participants and deliver the intervention first and then matched control group for reducing the drawback of non-randomization. Thus, the pretest was not administered at the same time to the control and experiential groups.The fourth limitation has added (p.14 line 528-533).
Consider changing the title of the “pretest” for the control group to clarify that it was not prior to the intervention rather reflects that it was 1st time administered and correlated with the 1st post test for the experimental group.
Response 1.3: The term of “pretest” has changed as “baseline” (p.1 line 27; p.3 line 296, 299, 301, 304, & 306; p.12 line 405, 412, 415, & 417; p.14 line 509).
The reviewer also has a question on the approach used to train the head nurses in the completion of the Spiritual Care Competence Scale (SCCS). Could their lack of understanding of the rating scale potentially influenced the results?
Response 1.4: We did not train the head nurses in the completion of the Spiritual Care Competence Scale (SCCS), because this SCCS has been used in our descriptive correlational study without misunderstanding for nurses to complete it. However, we have discussed the different finding between nurses and head nurses (p.11 line 390-398).
Tables are difficult to read with lack of separation of the content for different categories.
Response 1.5: Table A3 and A4 have revised (p.17).
Point 2: Specific comments
- Brief statement on matching process is needed as well as clarification of what the 1-3 ratio means.
Control group was matched for similar units, ages, working experience, and nursing clinical ladder with the experimental group (1:1-3 ratio).
Response 2.1: A sentence has added for explaining rationale of 1:1-3 (p.4 line 157-159). In order to have enough particpants of the control group and reduce the drawback of non-randomization, we use 1:1-3 ratio and the strategy of matching the experimental and control group for similar ages, working experience, and nursing clinical ladder. For example, 11F ward only has 15 beds and less nurses. It has more nurses participanted in the experimental group so that each participant of the experimental group help to find a participant with similar ages, working experience, and nursing clinical ladder for the control group from the same ward or division (hemotology and oncology unit). In addition, the sample size of the control group and the experimental group needs 68 respectively, but it is hard to recruit enough participants for the experiemtal group during COVID-19 pandemic.
- Table 1 is confusing. Be clear when the measures were administered to the control group and to the experimental group. Table 1 line 108
Table 1. Study design and measurements.
Response 2.2: Table 1 has revised (p.3 line 136).
- Clarify “longer and shorter periods of time. Line 116 plus
longer periods of time were included. Nurses and nursing managers who worked in the outpatient department, health checkups, emergency department, operating room, hemodialysis and peritoneal dialysis units, and chemotherapy room providing task oriented and shorter time care were excluded.
Response 2.3: The sentences regarding inclusion and exclusion criteria have revised again (p.4 line 149-153).
- Clarify - I believe the authors mean to indicate that there was no previously reported effect size in the literature Line 119
No previous effect size of between-subjects effects was used to estimate the sample size.
Response 24: The sentence of power analysis has revised and shows post hoc power analysis (p.4 line 166-173).
- How many times was the program offered? Recommend that the term “session” be replaced by program as session indicates to the reviewer that participants attended numerous times. Line 129
The scenario-based spiritual care course was about four hours each session, including a pretest, a lecture using
Response 2.5: The program offered 10 times from August 17, 2019 to November 28, 2020. The section of intervention has revised and add Table 2 (pp.5-6 line 219).
- Clarify what the 1-3 means in the statements below Line 143 plus I believe that the authors are indicating that there were 3 examiners and there was a total of 3 standardized patients
Examiners (n = 1-3) watched good and poor versions of a video two days before the OSCE examination for assessing inter-rater reliability. Standardized patients (n = 1-3) were qualified and trained before the OSCE examination.
Response 2.6: The number of examiners and standardized patients was deleted and add Table 2 for showing the number of participants each time. When we knew the number of registers of this program >11 in advance, the principal investigator (PI) would prepared 3 examiners, standardized patients, and examination rooms for OSCE one week before for shortening participants’ waiting time. However, registers of this program may not check in on the scheduled day so that the PI would inform examiners and standardized patients immediately for cancel event (p.5 line 209-211).
- Was it seven or six? Line 159
Then, seven or six experts on spiritual care practice, hospice and palliative care, and Buddhist chaplain in the clinical
Response 2.7: This study was belonged to a 2-year study, including a descriptive correlational study and an interventional study. So, the PI did content validity on SPS, SCPS-R, SCCS in September 2017 and reflective logs, SPFS, OSCE Checklist, & CSS in May 2019 respectively. Thus, the number of experts was 7 in September 2017 and 6 in May 2019. This sentence has revised (p.6 line 233-236).
- The amount of dollars distributed to the participants would be extreme for the organization I have been affiliated with. Are the amounts correct? Lines 172 to 181
500-dollar gift card in the afternoon. Experimental group participants helped to find nurses from the same units or division, with same age range, nursing clinical ladder, working years, and education who understood this study and were willing to fill out the spiritual care-related scales as a matched control group. After completing the scales and collecting them in a cardboard box within a week’s time, the control group participants were awarded a 200-dollar gift card. The head nurses of the experimental group received e-mails to explain the relevant information of this study and to obtain signed informed consent. The HN-evaluated SCCS was filled out and head nurses were given a 200-dollar gift card. Then, the SPS, SCPS-R, self- and HN-evaluated SCCS, and reflection log were collected from both groups three months after the in-181 intervention and were given a 200-dollar gift card.
Response 2.8: The 500-dollar gift card was New Taiwan dollars and was approved by the IRB. It is around 15 U.S. dollars. New Taiwan has added to those dollars (p.6 Table 2; p.7 line 258-260, 263-268, 270-274).
- Sentence edits - repetitive use of “more” in the section that follows. Unsure what the N4 means at the end of the section that follows- Line 202-204
Most nurses were single (73.2%) and more 202 than one-third of the nurses graduated from two-year colleges (43.5%). More than half 203 reported having a religion (58.7%). More than one-third of the nurses’ clinical ladder positions were N4 (36.2%).
Response 2.9: This sentence has revised (p.8 line 295).
- Table 3 - recommend that the table be formatted so that the variables stand out. Currently the variables are in the same list as the variable options.
Response 2.10: Table 3 of the original manuscript has standed out variables and categories by aligning left and indent, but the editor has aligned center. Thus, it is not easy to variables and their categories. Table 3 has revised for adding category of the variables (p.9 line 301).
- In table 2 one of the options for Religion is No - consider changing to “None identified”
Religion No
Response 2.11: The term of “no” has changed as “none identified” (p.9).
- In table 2 the term “Nursing Clinical Ladder” is used as a variable header. What do the options ries mean?
Nursing clinical ladder
Response 2.12: Nursing clinical ladder is a categorical variable and only show frequency and percent in the Table 2 (p.9).
- “Had no - change wording to “was not” Line 234
Additionally, the experimental group had no significantly higher mean HN-evaluated SCCS scores at the 2nd posttest than the control group after adjusting for covariates (102.3915.76 vs. 94.2314.74, respectively; b = -0.98, 236 p = .695).
Response 2.13: The term has changed (p.10 line 342).
- Results are presented logically. One thing that would be helpful is to note the meaning of the scale results as not all of the scales have the same direction (higher rating reflects higher spiritual competence, etc). It is on the measures table but telling the reader this in the results section would be helpful.
Response 2.14: Thanks for the comments. Notes of Table 4, A2, A3, and A4 have added “Higher SPS or SCCS scores indicate a greater spiritual perspective or spiritual care competence; Lower SCPS-R scores indicate a positive spiritual care perspective” (p.10 line 327-329, p.18 line 638-639, 643-645; p.19 linw 649-650).

Reviewer 2 Report
Introduction
The authors seem to group spirituality and religious belief as one and the same. It is well accepted in current research that the concept of spirituality is much broader than organised religion. This is one of the reasons why the American Nurses Association separates religious and spiritual beliefs (line59-61). While religion and spirituality are similar in its foundation, they are very different in practice. While religion is an organised community-based system of beliefs, spirituality is something that resides inside the individual and based on personal beliefs (see for example Rogers and Wattis (2015). Spirituality in nursing practice. Nursing Standard, 29(39), pp. 51-57. DOI:10.7748/ns.29.39.51.e9726. This is a major shortcoming as the authors is providing an oversimplistic description of the relationship between enhancement in care and religious belief, while paying tokenistic attention to spirituality.
Rows 51-53.
The statement that religious beliefs is associated with…. Is in part incorrect in that it is both religious belief and spirituality that leads to positive outcomes, a notion supported by the reference the authors have chosen to support this argument.
Reference list
Reference list needs to be amended as the numbering is doubling up.
Author Response
Point 1: Introduction
The authors seem to group spirituality and religious belief as one and the same. It is well accepted in current research that the concept of spirituality is much broader than organised religion. This is one of the reasons why the American Nurses Association separates religious and spiritual beliefs (line 59-61). While religion and spirituality are similar in its foundation, they are very different in practice. While religion is an organised community-based system of beliefs, spirituality is something that resides inside the individual and based on personal beliefs (see for example Rogers and Wattis (2015). Spirituality in nursing practice. Nursing Standard, 29(39), pp. 51-57. DOI:10.7748/ns.29.39.51.e9726. This is a major shortcoming as the authors is providing an oversimplistic description of the relationship between enhancement in care and religious belief, while paying tokenistic attention to spirituality.
Response 1: We did not group spirituality and religious belief as one and the same. We totally agree with the 2nd sentence to the 5th sentence. We found that clinical nurses usually confused with spirituality and religion and do not know how to assess patients’ spirituality. That is why we conduct this study to educate nurses about the concept of spirituality and religion and other concepts of spirituality. The 1st learning objective was to differentiate the definitions of spirituality and religion. Please see the detailed intervention of Table 2 (p.5).
Point 2: Rows 51-53.
The statement that religious beliefs is associated with…. Is in part incorrect in that it is both religious belief and spirituality that leads to positive outcomes, a notion supported by the reference the authors have chosen to support this argument.
Response 2: This sentence and citation has revised (p.2 line 61-63).
Point 3: Reference list
Reference list needs to be amended as the numbering is doubling up.
Response 3: Our submitted manuscript was not doubling up numbering of reference list. Yet, the editor has revised it and became doubling up numbers of reference list. All doubling up numbering of reference list has deleted (pp.20-23).

Reviewer 3 Report
Manuscript ID: healthcare-2070427, entitled “The Effects of a Scenario-Based Spiritual Care Course on Spiritual Care Competence among Clinical Nurses: A Quasi-Experimental Study”
I wish the thank the editors for the opportunity to review “The Effects of a Scenario-Based Spiritual Care Course on Spiritual Care Competence among Clinical Nurses: A Quasi-Experimental Study”. I applaud the authors for their work, however, there are some issues needed to be clarified in the presentation in this research.
1. Title [The Effects of a Scenario-Based Spiritual Care Course on Spiritual Care Competence among Clinical Nurses: A Quasi-Experi-3 mental Study]. Please confirm whether it is suitable to revise to [The Effects of a Scenario-Based Spiritual Care Course on Spiritual Care Competence among Clinical Nurses: A Quasi-Experi-3 mental Study].
2. Abstract: This study had pre-test, post-test immediately, and post-test at 3 month after intervention. Please mention in the Abstract.
3. 2.1.Study Design, the second paragraph should be merged into Introduction. In addition, the authors mentioned (1:1-3ratio), this maybe remove into “2.2. Participants and Setting”.
4. Table 1., Note should include all abbreviations, such as NR, O1, O2, O3.
5. 2.3. Intervention, “The content validity of the scenario, tem-148 plate, and checklist, and SP feedback scale were reviewed by six experts using a 4-point 149 Likert scale.” How about the CVI?
6. Table 3, how to differentiate low competence, moderate competence, and high competence? It should be mentioned in Table A1 or content?
7. Figure1 could follow CONSORT FLOWDIAGRAM to make clear. Please revised it.
8. Table 2. Homogenity test…, Some cells were small sample size. Maybe some categories could be merged.
9. In conclusion, “Future studies and in-service education may support spiritual care competence by cultivating clinical nurse preceptors with better spiritual care pedagogy through the use of entrustable professional activities (EPAs) for bedside teaching during nurses’ postgraduate years”. The authors mentioned about EPAs. Please confirm it.
Author Response
I wish the thank the editors for the opportunity to review “The Effects of a Scenario-Based Spiritual Care Course on Spiritual Care Competence among Clinical Nurses: A Quasi-Experimental Study”. I applaud the authors for their work, however, there are some issues needed to be clarified in the presentation in this research.
Point 1: Title [The Effects of a Scenario-Based Spiritual Care Course on Spiritual Care Competence among Clinical Nurses: A Quasi-Experimental Study]. Please confirm whether it is suitable to revise to [The Effects of a Scenario-Based Spiritual Care Course on Spiritual Care Competence among Clinical Nurses: A Quasi-Experimental Study].
Response 1: The suggestion is similar to our title.
Point 2: Abstract: This study had pre-test, post-test immediately, and post-test at 3 month after intervention. Please mention in the Abstract.
Response 2: We have added to the abstract (p.1 line 29-30).
Point 3: 2.1.Study Design, the second paragraph should be merged into Introduction. In addition, the authors mentioned (1:1-3ratio), this maybe remove into “2.2. Participants and Setting”.
Response 3: The second paragaph of the study design has moved to the introduction (p.3 line 113-129) and 1:1-3 ratio has added to the section of participants and setting (p.4 line 136-159).
Point 4: Table 1., Note should include all abbreviations, such as NR, O1, O2, O3.
Response 4: All abbreviations have added to note of the Table 1 (p.3 line 136).
Point 5: 2.3. Intervention, “The content validity of the scenario, template, and checklist, and SP feedback scale were reviewed by six experts using a 4-point 149 Likert scale.” How about the CVI?
Response 5: The CVI (1.00) of the scenario and template of the OSCE has added to the last sentence (pp.4-5 line 187-190). The CVI of OSCE checklist and SP feedback scale shows in the Table A1 (p.17).
Point 6: Table 4, how to differentiate low competence, moderate competence, and high competence? It should be mentioned in Table A1 or content?
Response 6: The SCCS has been categorized as low (<64), moderate (64-98), and high competence (>98) shows in Table A1. However, there were no detailed information on cited references about how did the authors differentiate low, moderate, and high competence.
Point 7: Figure1 could follow CONSORT FLOWDIAGRAM to make clear. Please revised it.
Response 7: Figure 1 has revised using CONSORT flow diagram (p.8).
Point 8:Table 3. Homogenity test…, Some cells were small sample size. Maybe some categories could be merged.
Response 8: Education and religion of the Table 3 has recategorized again, but 2-year/5-year diploma or I-Kuan Tao cannot be collapsed according to different educational system and type of religion in Taiwan (p.9 line 301).
Point 9: In conclusion, “Future studies and in-service education may support spiritual care competence by cultivating clinical nurse preceptors with better spiritual care pedagogy through the use of entrustable professional activities (EPAs) for bedside teaching during nurses’ postgraduate years”. The authors mentioned about EPAs. Please confirm it.
Response 9: The studied hospital and other hospitals in Taiwan have held educational training of entrustable professional activities (EPAs) on clinical nurse preceptors for applying EPAs to clinical nursing education on nurses’ postgraduate years. Now, they only focus on basic competence of clinical patient care. A full EPA description for spiritual care competency (title, specification and limitations, risks in failure, relevant competency domains, required knowledge, skills, and attitude and experiences, before entrustment, sources of information for assessment, level and expected moment of entrustment, and expiration) on clinical nurses by different nursing clinical ladder can be designed in the future.

Reviewer 4 Report
Reviewer comments, manuscript “The Effects of a Scenario-Based Spiritual Care Course on Spiritual Care Competence among Clinical Nurses: A Quasi-Experimental Study”, Healthcare
In my view, the strengths of this paper are its innovative topic and the multi-method evaluation approach. Generally, the presentation is detailed. However, some relevant information is lacking or presented in a way that is rather difficult to follow (description of the intervention, OSCE, discussion of results). My comments and suggestions to address these aspects are as follows:
Introduction
p. 2, line 49: What do the “spiritual needs and issues” comprise? Some examples would be helpful here.
p. 2, lines 55 et seq.: Similar points apply here. The explanations regarding spiritual care and its importance would be better understood if some examples were given, e.g., what do spiritual competencies or a spiritual care plan include?
p. 2, lines 64 et seq.: Are there any data on how important relevant nurses themselves rate spiritual issues in their clinical practice? This would be interesting regarding the relevance of these topics in nursing education (or might help explain why there are few educational approaches).
Method
Study design
p. 3, table 1: There are some abbreviations in this table that are not explained in the note (NR, O1-3, HN(-evaluated)). Please add an explanation.
Intervention
p. 3, lines 129 et seq.: To me, in its current description, the intervention is a bit difficult to understand. Is it correct that the course was 4 hours in total? How many sessions were in one course (which was described as a one-day course above)? Or do sessions correspond to cohorts (i.e., the course was conducted ten times)? If so, the term “session” seems a bit misleading. A revision of this passage would be helpful.
p. 3, lines 124: On what basis were the teaching/learning goals set? Were they derived from the literature, from empirical findings, from the abovementioned nursing standards, …? This information should be added.
p. 4, lines 142 et seq.: Were there other OSCE stations? And was the spiritual care station developed specifically for this study?
Table A.1: Please check the layout of the table. Some of the text in the “operational definition” column seems to have shifted or does not match the text in the previous column (e.g., Spiritual Care Perspective Scale-Revised (SCPS-R): “10 items of 5-point Likert scale…”). Additionally, please add the year of publication of Hsieh in the “OSCE Checklist” row.
Data collection
p. 4, lines 172 et seq.: It might be useful to add “New Taiwan Dollar” here.
p. 4, lines 172 et seq., and figure 1: When were the control group participants recruited? The way the information is currently presented, one might get the impression that recruitment took place consecutively after the experimental group participants had completed their assessment.
Results
p. 6, table 2: The clinical ladder levels (N1 etc.) are probably unfamiliar to some readers and should be briefly explained.
p. 8, lines 273 et seq.: Most checklist items given seem to refer to some general interpersonal and communicative skills (e.g., eye contact, listening behavior, encouraging patients to express emotions). How many items (and thus assessment criteria) were specifically referring to spiritual care-related aspects/skills? To what extent can this OSCE assessment provide information on spiritual care competence (in terms of content validity)? This issue should also be addressed and discussed in the discussion section (p. 10, lines 398 et seq.).
I would suggest including brief summaries throughout the results section (given the high “density” of information), which would help readers process the results.
Discussion
p. 9, lines 346 et seq.: Another aspect that might play a role here is how well spiritual care competencies can actually be observed or assessed by others, as they are highly subjective.
p. 9, lines 361/362: Since this was a non-randomized study, the authors’ conclusion that the increase in spiritual care skills “is attributed to the effects of the (…) program” should be phrased more cautiously.
p. 11, lines 418 et seq.: The discussion regarding SP feedback is, in my view, a bit confusing. Does the statement “The SP indicates experimental group nurses’ better performance as “the examinee spends time with me to talk”” etc. indicate that nurses with higher OSCE/performance scores were rated more positively in this area, …?
Generally, the discussion section seems a bit too detailed and small-scale, making the information in part difficult to follow. It could thus be streamlined. Moreover, the reference to existing evidence and concepts is somewhat neglected.
Minor comments
Results
p. 7, line 246: Please delete “the” (“…significantly lower than the that at the…”).
p. 8, line 294: Please write “groups” instead of “group”.
References
In reference no. 22, please write “dissertation” instead of “dissertations”.
Author Response
In my view, the strengths of this paper are its innovative topic and the multi-method evaluation approach. Generally, the presentation is detailed. However, some relevant information is lacking or presented in a way that is rather difficult to follow (description of the intervention, OSCE, discussion of results). My comments and suggestions to address these aspects are as follows:
Point 1.1: Introduction
- 2, line 49: What do the “spiritual needs and issues” comprise? Some examples would be helpful here.
Response 1.1: Some examples of spiritual needs and issues have added (p.2 line 50-53).
Point 1.2: p. 2, lines 55 et seq.: Similar points apply here. The explanations regarding spiritual care and its importance would be better understood if some examples were given, e.g., what do spiritual competencies or a spiritual care plan include?
Response 1.2: p.2 line 56-57 indicates the importance of spiritual care as “Evidence has shown that spirituality is associated with better physical and mental health”. P.2 line 61-63 points out 6 dimensions of nursing competence for spiritual care.
Point 1.3: p. 2, lines 64 et seq.: Are there any data on how important relevant nurses themselves rate spiritual issues in their clinical practice? This would be interesting regarding the relevance of these topics in nursing education (or might help explain why there are few educational approaches).
Response 1.3: The SCPS-R has 2 questions related this comment. For instance, Q1. Spiritual care is a significant part of nursing practice. Q3.The domain of nursing practice should include spiritual care. Mean and SD of both questions were across 3-time point respective: Q1 2.07±0.72 (T1), 1.66±0.59 (T2), 1.94±0.67 (T3) and Q3 2.08±0.72 (T1), 1.58±0.63 (T2), 1.97±0.70 (T3). After receiving this program, average of both questions slightly decrease (positive perspective). Yet, after 3 months, nurses’ averages slightly increase again. In addition, Green & Kim-Godwin (2020) found 81.7% of freshman of the postlicence program believed that addressing patients’ spiritual needs within the role of the professional nurse. A strong association between receiving spiritual care education in prelicensure programs and self-reported feelings of preparedness (p<.001), as well as between receiving spiritual care education at work and self-reported feelings of preparedness (p=.005). We have added several citations’ information (p.2 line 75-77).
Point 2.1: Method
Study design
- 3, table 1: There are some abbreviations in this table that are not explained in the note (NR, O1-3, HN(-evaluated)). Please add an explanation.
Response 2.1: All abbreviations of the Table 1 has added (p.3 line 136).
Point 2.2: Intervention
- 3, lines 129 et seq.: To me, in its current description, the intervention is a bit difficult to understand. Is it correct that the course was 4 hours in total? How many sessions were in one course (which was described as a one-day course above)? Or do sessions correspond to cohorts (i.e., the course was conducted ten times)? If so, the term “session” seems a bit misleading. A revision of this passage would be helpful.
Response 2.2: The section of intervention has revised and added Table 2 (pp.4-6 line 174-216 & 219).
Point 2.3: p. 3, lines 124: On what basis were the teaching/learning goals set? Were they derived from the literature, from empirical findings, from the abovementioned nursing standards, …? This information should be added.
Response 2.3: The sentence of learning objectives has revised and added citations (p.4 line 175-179).
Point 2.4: p. 4, lines 142 et seq.: Were there other OSCE stations? And was the spiritual care station developed specifically for this study?
Response 2.4: We borrowed the OSCE examination rooms of the clinical skill center from the studied hosptial. Please see place of the Table 2 (pp.4-5 line 201-202 & 219).
Point 2.5: Table A.1: Please check the layout of the table. Some of the text in the “operational definition” column seems to have shifted or does not match the text in the previous column (e.g., Spiritual Care Perspective Scale-Revised (SCPS-R): “10 items of 5-point Likert scale…”). Additionally, please add the year of publication of Hsieh in the “OSCE Checklist” row.
Response 2.5: The editor has alighned center 5 columns so that it is not easy to read Table A1. Horizontal lines have added this table and shorten space on each bullet for easy to read this table (pp.16-17).
Point 2.6: Data collection
- 4, lines 172 et seq.: It might be useful to add “New Taiwan Dollar” here.
Response 2.6: New Taiwan Dollar has added to the section of data collection (p.7 line 240-274).
Point 2.7: p. 4, lines 172 et seq., and figure 1: When were the control group participants recruited? The way the information is currently presented, one might get the impression that recruitment took place consecutively after the experimental group participants had completed their assessment.
Response 2.7: This study recruited participants of the experimental group and measure outcomes before (baseline) and after (1st posttest) the educational program. Then, the participants of the experimental group help to find the matched control group and measure outcomes (baseline). Figure 1 (p.8) and Table 1 (p.3 line 136) have revised.
Point 3.1: Results
- 6, table 3: The clinical ladder levels (N1 etc.) are probably unfamiliar to some readers and should be briefly explained.
Response 3.1: We have added a note of clinical ladder system in Taiwan below the table (p.9 line 303-311).
Point 3.2: p. 8, lines 273 et seq.: Most checklist items given seem to refer to some general interpersonal and communicative skills (e.g., eye contact, listening behavior, encouraging patients to express emotions). How many items (and thus assessment criteria) were specifically referring to spiritual care-related aspects/skills? To what extent can this OSCE assessment provide information on spiritual care competence (in terms of content validity)? This issue should also be addressed and discussed in the discussion section (p. 10, lines 398 et seq.).
Response 3.2: According to the OSCE scenario and template, the task was to do spiritual assessment with 15 minutes for the 1st admitted high-grade cervical cancer girl. Thus, the principal investigator (PI) developed this checklist through literature review (van Leeuwen et al., 2009; van Leeuwen & Schep-Akkerman, 2015; McSherry, 2006; Hsiao et al., 2009; Ku, 2010) and personal clinical experience. 6 experts of 1 social worker, 1 head nurse, and 1 physician of hospice and palliative care, 1 nursing professor (Christian), and a head nurse of teaching and research division of the nursing department. They are familiar with spiritual assessment and OSCE. The CVI was 0.94 (Table A1 p.17). This study’s Cronbach alpha was 0.84. The principal axis factoring with varimax rotation of the exploratory factor analysis showed two factor (factor 1: Q2-5, Q7-8 and factor 2: Q5.Can respond appropriately to patient questions & Q11.Can assess relevant data of the spiritual aspect). We have added a sentence here (p.14 line 515-516).
Point 3.3: I would suggest including brief summaries throughout the results section (given the high “density” of information), which would help readers process the results.
Response 3.3: A summary has added to the last paragraph of the results (p.12 line 432-440).
Point 4.1: Discussion
- 9, lines 346 et seq.: Another aspect that might play a role here is how well spiritual care competencies can actually be observed or assessed by others, as they are highly subjective.
Response 4.1: A sentence has added to here (p.13 line 436-439).
Point 4.2: p. 9, lines 361/362: Since this was a non-randomized study, the authors’ conclusion that the increase in spiritual care skills “is attributed to the effects of the (…) program” should be phrased more cautiously.
Response 4.2: The sentence has revised (p.13 line 472).
Point 4.3: p. 11, lines 418 et seq.: The discussion regarding SP feedback is, in my view, a bit confusing. Does the statement “The SP indicates experimental group nurses’ better performance as “the examinee spends time with me to talk”” etc. indicate that nurses with higher OSCE/performance scores were rated more positively in this area, …?
Response 4.3: These sentence has revised (p.14 line 539-552).
Point 4.4: Generally, the discussion section seems a bit too detailed and small-scale, making the information in part difficult to follow. It could thus be streamlined. Moreover, the reference to existing evidence and concepts is somewhat neglected.
Response 4.4: The discussion section has reviewed and revised (pp.12-15 line 442-590).
Point 5.1: Minor comments
Results
- 7, line 246: Please delete “the” (“…significantly lower than the that at the…”).
Response 5.1: We has deleted “the” (p.10 line 336).
Point 5.2: p. 8, line 294: Please write “groups” instead of “group”.
Response 5.2: We has revised it as “groups”.
Point 5.3: References
In reference no. 22, please write “dissertation” instead of “dissertations”.
Response 5.3: We has revised it (no.35) as “dissertation” (p.21 line 738).

Reviewer 5 Report
Spirituality is a significant issue for many people, especially those in vulnerable situations, such as frail older adults. As the authors state, spirituality is the essence of existence and provides meaning and purpose in one’s personal existence.
This article was written in this direction managed to examine and discuss this important issue in a well written and analysed manner.
The discussion section is thoroughly written providing the readers a well-documented picture of the topic under study.
Who provided the gift vouchers?
What is the total number of participants in the study?
Author Response
Point 1: Spirituality is a significant issue for many people, especially those in vulnerable situations, such as frail older adults. As the authors state, spirituality is the essence of existence and provides meaning and purpose in one’s personal existence.
Response 1: Thanks for the feedback.
Point 2: This article was written in this direction managed to examine and discuss this important issue in a well written and analysed manner.
Response 2: Thanks for the feedback.
Point 3: The discussion section is thoroughly written providing the readers a well-documented picture of the topic under study.
Response 3: Thanks for the feedback.
Point 4: Who provided the gift vouchers?
Response 4: The section of data collection has added the providers of the gift vouchers (p.7 line 249-274). The gift vouchers of the experimental group at the 1st posttest were provided by the principal investigator after OSCE, while the gift vouchers of the experimental group at the 2nd posttest were provided by the research assistant via internal envelopes. The gift vouchers of the control group and the head nurses at the baseline and the posttest were provided by the research assistant via internal envelopes.
Point 5: What is the total number of participants in the study?
Response 5: The total number of participants in this study was 138 (Table 3 & 4 pp.9 & 10).

Reviewer 6 Report
Revision of the article:
The Effects of a Scenario-Based Spiritual Care Course on Spiritual Care Competence among Clinical Nurses: A Quasi-Experimental Study
This article aims to validate the effects of a scenario-based spiritual care course on spiritual care competence among clinical nurses.
This is an article with a very new and interesting theme. It is a well-designed study, despite the fact that, as the authors point out in the limitations section, the study lacks randomization and the sample of the control group is higher than the sample of the experimental group.
It is a publishable article that brings new information on how to develop spiritual competence in nurses. However, prior to its publication it is necessary to correct some existing failures in the references:
· When several references are cited in a row with consecutive numbers, only the first and last reference numbers should be written. In this sense, it is important to correct the references of paragraphs 64, 70 and 76.
· In the final section of bibliographical references, the reference numbers are repeated twice and the numbers of references 48 to 51 do not match.
· Please review reference number 27, 38, 39 and 40. Data are missing, or they are not fully referenced.
Review the instructions on how to reference bibliographic citations at the following link please: https://www.mdpi.com/journal/healthcare/instructions
Kind regards
Author Response
Revision of the article:
The Effects of a Scenario-Based Spiritual Care Course on Spiritual Care Competence among Clinical Nurses: A Quasi-Experimental Study
Point 1: This article aims to validate the effects of a scenario-based spiritual care course on spiritual care competence among clinical nurses. This is an article with a very new and interesting theme. It is a well-designed study, despite the fact that, as the authors point out in the limitations section, the study lacks randomization and the sample of the control group is higher than the sample of the experimental group.
Response 1: Thanks for the feedback.
Point 2:It is a publishable article that brings new information on how to develop spiritual competence in nurses. However, prior to its publication it is necessary to correct some existing failures in the references:
When several references are cited in a row with consecutive numbers, only the first and last reference numbers should be written. In this sense, it is important to correct the references of paragraphs 64, 70 and 76.
In the final section of bibliographical references, the reference numbers are repeated twice and the numbers of references 48 to 51 do not match.
Please review reference number 27, 38, 39 and 40. Data are missing, or they are not fully referenced.
Review the instructions on how to reference bibliographic citations at the following link please: https://www.mdpi.com/journal/healthcare/instructions
Response 2: All citations and references have checked and revised again (pp.2-23).

Round 2
Reviewer 2 Report
With the amendment made to the manuscript I am now happy to recommend it for publication. The introduction now provides a clearer distinction between religious belief and spirituality which improve the overall quality of the article, which also corresponds better to what the authors have written in their findings.
Author Response
Point 1: With the amendment made to the manuscript I am now happy to recommend it for publication. The introduction now provides a clearer distinction between religious belief and spirituality which improve the overall quality of the article, which also corresponds better to what the authors have written in their findings.
Response 1: Thanks for the comments and encouragement.

Reviewer 4 Report
I would like to thank the authors for their thorough revision of the manuscript which has, in my view, gained a lot from the revisions. My comments and suggestions have been adequately addressed.
Two minor suggestions:
- Line 76: Please write „show“ instead of „shows“.
- Lines 431-433: The mean values are not very meaningful without indicating the range of values (minimum, maximum) in which they fall. Please add this information here.
After these items have been taken care of, I would recommend acceptance of the manuscript.
Author Response
Point 1: I would like to thank the authors for their thorough revision of the manuscript which has, in my view, gained a lot from the revisions. My comments and suggestions have been adequately addressed.
Response 1: Thanks for the comments and encouragement.
Point 2: Two minor suggestions:
- Line 76: Please write „show“ instead of „shows“.
- Lines 431-433: The mean values are not very meaningful without indicating the range of values (minimum, maximum) in which they fall. Please add this information here.
After these items have been taken care of, I would recommend acceptance of the manuscript.
Response 2: Line 76 has changed as show (p.2). Line 436-437 have added the range of values (p.12). Thanks for the comments and encouragement.
